# Cereal β-d-Glucans in Food Processing Applications and Nanotechnology Research

**DOI:** 10.3390/foods13030500

**Published:** 2024-02-04

**Authors:** Lucie Jurkaninová, Václav Dvořáček, Veronika Gregusová, Michaela Havrlentová

**Affiliations:** 1Department of Food Science, Faculty of Agrobiology, Food and Natural Resources, Czech University of Life Sciences, Kamýcká 129, 165 00 Praha, Czech Republic; jurkaninova@af.czu.cz; 2Crop Research Institute, Drnovská 507, 161 06 Prague, Czech Republic; dvoracek@vurv.cz; 3Department of Biotechnology, Faculty of Natural Sciences, University of Ss. Cyril and Methodius, Námestie J. Herdu 2, 917 01 Trnava, Slovakia; veronika.gregusova93@gmail.com; 4National Agricultural and Food Center—Research Institute of Plant Production, Bratislavská Cesta 122, 921 68 Piešťany, Slovakia

**Keywords:** β-d-glucans, cereals, extraction, processing, food application, nanotechnology

## Abstract

Cereal (1,3)(1,4)-β-d-glucans, known as β-d-glucans, are cell wall polysaccharides observed in selected plants of grasses, and oats and barley are their good natural sources. Thanks to their physicochemical properties β-d-glucans have therapeutic and nutritional potential and a specific place for their functional characteristics in diverse food formulations. They can function as thickeners, stabilizers, emulsifiers, and textural and gelation agents in beverages, bakery, meat, and extruded products. The objective of this review is to describe the primary procedures for the production of β-d-glucans from cereal grains, to define the processing factors influencing their properties, and to summarize their current use in the production of novel cereal-based foods. Additionally, the study delves into the utilization of β-d-glucans in the rapidly evolving field of nanotechnology, exploring potential applications within this technological realm.

## 1. Introduction

Generally, β-glucans are considered as dietary fibers and a complex of polysaccharides, which are derived from various sources from selected groups of plants, algae, fungi, or microorganisms [1]. In the global market, the sales of plant and mushroom β-glucans reached USD 410.6 million in 2020. The market is expected to experience a compound annual growth rate of 7.6%, reaching an estimated sales value of USD 476.5 million by the end of 2025 [2].

The cereal (1,3)(1,4)-β-d-glucans, also known as β-d-glucans, are polysaccharides predominantly found in the cell walls of plants belonging to grasses [3]. Cereal β-d-glucans, particularly those from oats and barley, possess specific molecular-structural characteristics that distinguish them in molecular weight (MW), degree of branching, and linkage types from the next important β-d-glucan sources, such as fungi and yeasts [2,4]. Cereal β-d-glucans are polysaccharides consisting of linear chains of D-glucopyranosyl residues connected by a combination of β-(1,4) and β-(1,3) linkages [5] (see Figure 1).

The degree of polymerization (DP) of common β-d-glucans in grasses is about 1000 or more and is unique to each cereal. The physicochemical properties of β-d-glucans such as MW, basic structure, chain length, branching, and solubility influence their biological activities and determine their applications in tissue engineering, food, biomedical, pharmaceutical, and cosmetic industries [7]. 

The β-d-glucans content in cereals varies across different grains, with wheat grains typically containing around 1%, oats ranging from 3% to 7%, which is primarily found in the endosperm cell wall (approximately 75%), and bran (around 10.4%), 82% of which is soluble. Barley exhibits a content of 5% to 11% [8,9]. Non-starch polysaccharides including β-d-glucans, recognized as a valuable source of dietary fiber, have garnered significant interest due to their potential to combat emerging diseases. Extensive research, both in vitro and in vivo, has documented their notable health benefits, establishing their role as antioxidants, antitumor agents, anti-inflammatory agents, antimicrobial agents, and antidiabetic agents [9,10,11]. The health requirements for β-d-glucans from oats and barley are justified in terms of maintaining normal blood low-density lipoprotein (LDL) cholesterol levels, increasing satiety leading to lower energy intake, and reducing postprandial glycemic response and digestive functions [6,12]. The approved health claims in terms of lowering glucose levels in the blood serum are observed when at least 4 g of β-d-glucans, from oats or barley, per 30 g of available carbohydrates is consumed in a meal [12]. It is also proven that high-molecular-weight (HMW) oat β-d-glucans, consumed at 3–4 g per day in solid foods, reduce LDL cholesterol [13]. Intake of cereal β-d-glucans or their natural sources in appropriate and approved quantities is safe and brings proven health benefits for the consumer.

The cereal polysaccharides have also acquired a specific place for their functional characteristics in diverse food formulations. They can function as thickeners, stabilizers, emulsifiers, textural agents, and gelation agents in milk and milk products, bakery products, meat products, and extruded products [14,15,16]. Recently, the therapeutic potential of predominately fungal β-glucans has been further expanded using nanotechnological approaches [17]. The objective of this review is to outline the primary procedures for isolating cereal β-d-glucans, to define the processing factors influencing their properties, and to describe their current use in the production of cereal-based foods. Additionally, the study delves into the utilization of β-d-glucans in the rapidly evolving field of nanotechnology, exploring potential applications within this technological realm.

## 2. Production of β-d-Glucans from Cereal Grains

Commonly used methods for β-d-glucan extraction from cereals on an industrial scale include hot water, alkali, enzyme (as elicitor), solvent, and superheated water [18]. The typical extraction process of cereal β-d-glucans includes three main steps (pre-processing, extraction, and isolation). The schematic for β-d-glucan production under laboratory or pilot plant conditions is presented in Figure 2. 

### 2.1. Extraction Procedures for β-d-Glucans from Cereals 

The recovery of β-d-glucans from the cell walls of endosperm cell grains (e.g., barley and oats), where they coexist with starch, matrix protein, and lipid reserves, poses challenges due to their complex structure and overlapping solubility. The extraction procedures employed in studying the physicochemical properties of isolated β-d-glucans fractions necessitate optimization of yield, purity, and preservation of the β-d-glucans molecule’s integrity. However, achieving these objectives requires a delicate balance and compromise. High viscosity is a hallmark of β-d-glucans and is a function of the concentration of β-d-glucans in solution and their MW [6].

Processing can potentially alter the molecular composition (including the chemical structure and degree of polymerization), structural characteristics (such as molecular interactions), and functional attributes (such as viscosity, water-binding capacity, and solubility) of β-d-glucans. Consequently, extraction procedures have the potential to impact the sensory perception, physiological effects, and ultimately the health advantages associated with β-d-glucans [3,20,21,22].

Extraction and separation of β-d-glucans have garnered attention in various food industries as they offer potential applications as functional food ingredients or quality improvers [23]. Cereal β-d-glucans have distinct physicochemical properties and are more expensive to extract compared to β-glucans from other sources [24]. Two primary extraction procedures have been developed: dry extraction and wet extraction. These methods, as outlined by Brennan and Cleary [20], are designed to preserve the integrity of β-d-glucan molecules while optimizing their yield and purity.

The dry extraction involves milling and sieving to separate and concentrate grain components rich in β-d-glucans, which are then directly incorporated during the processing of food products. However, this method yields a relatively low product yield, usually less than 20% [23,25]. To enhance the concentration of naturally occurring β-d-glucans, Sibakov et al. [23] employed ultrafine grinding and electrostatic separation techniques, achieving a final product with a content of 56.2% β-d-glucans. 

The wet extraction methods involve inactivating endogenous enzymes and extracting β-d-glucans using water or alkaline solutions. Contaminants such as proteins and starch can be removed through hydrolytic enzyme treatment or selective adsorption. Alcohol precipitation is used for β-d-glucans separation, followed by drying through freeze-drying or spray drying [26,27]. Controls of temperature, pH, extraction time, and particle size are crucial in wet extraction methods [23].

Wet extractions can achieve higher yields (>50%) compared to dry extractions, but the high viscosity of aqueous extracts and the subsequent drying process result in increased costs. Moreover, low pH and endogenous enzymes may reduce the molar mass of β-d-glucans when exposed to moisture [23,28]. Subsequent enzymatic treatments with α-amylase and protease are commonly used for β-d-glucans purification [27]. Multiple enzymatic combinations can be used to reduce contaminants during the extraction process [29].

The prevailing method for wet extraction of β-d-glucans from cereals involves the utilization of hot water. Other wet methods include alkali extraction, acidic extraction, ultrasound-assisted extraction, microwave-assisted extraction, or enzyme-assisted extraction [15,30,31,32,33,34]. However, some of these methods have drawbacks such as long extraction time, high process costs, and reduced environmental sustainability [35]. Various critical factors affect the yield and properties of extracted β-d-glucans for use in food formulation. These factors include the type of solvent, temperature, pH, extraction time, agitation, particle size, and lipid content [9]. Each factor plays a crucial role in determining the extraction efficiency and properties of the extracted β-d-glucans, and the choice of extraction procedure significantly affects the structure and MW of the polysaccharide [19,21,36]. 

### 2.2. Separation Techniques Used in β-d-Glucans Production from Cereals 

Modern separation techniques include high-pressure extraction and supercritical fluid extraction. For instance, pressurized hot water extraction under elevated conditions (155 °C, 18 min, 50 bar) has demonstrated a reduction in extraction time and an increase in MW compared to conventional processes. Pressurized hot water extraction at 20 bar and 155 °C for 105 min yielded 97% purified β-d-glucans with an approximate molar mass of 500 kDa, demonstrating the high efficiency of this method [25]. 

Supercritical fluid extraction of β-d-glucans has shown promising results, producing β-d-glucan gels with more consistent structure than compared to freeze-dried methods [37]. Subcritical-water extraction under high temperature and pressure conditions (200 °C, 10 min) has also been explored as a single-step process for β-d-glucans extraction, which doubled the yield of this polysaccharide compared to using 60 °C warm water for 3 h [31,38]. 

Chemical extraction methods have also been explored to improve β-d-glucans yields. Alkaline extraction, acid extraction, dimethyl sulfoxide (DMSO), and urea have been employed. Alkaline extraction using NaOH and Na_2_CO_3_, ammonium sulfate, 2-propanol, or ethanol has been shown to inactivate β-glucanase enzymes responsible for yield reduction and result in increased MW and viscosity of the extracts [34]. Acidic extraction, although less studied, has been reported to denature endogenous hydrolytic enzymes and increase β-d-glucans solubility [28,39]. Du et al. [31] published an extraction method for β-d-glucans from hull-less barley bran using accelerated solvent extraction, and Vasanthan and Temelli [40] developed a process using organic solvents, acidified water, and aqueous alkali to concentrate β-d-glucans from barley and oat grains.

### 2.3. Purification of β-d-Glucans from the Cereal Matrix 

Purification methods, such as enzymatic hydrolysis and repeated precipitations of residual starch, are generally used for research purposes, achieving high purity levels of β-d-glucans [41,42,43]. Moreover, the increasing popularity of these enzymatic extractions can be attributed to their environmentally friendly nature. Studies have further indicated that enzymatic extraction yields were higher compared to alkaline and acidic extractions [44]. Enzymatic extraction processes can also remove starch, fat, and pentosans more effectively. The order of MW and yield of β-d-glucans based on different extraction procedures, according to Babu [45], is as follows: enzymatic > acidic > alkaline. The type and concentration of the used enzyme can influence both the MW and yields of β-d-glucans. Additionally, the extraction methods may affect the color of β-d-glucans, which can impact the properties of the final product.

It can be summarized that the operating conditions for β-d-glucans extraction, such as solvent type, temperature, pH, extraction time, and agitation, are crucial factors. Other factors that influence the extraction efficiency, chemical structure, and properties of isolated β-d-glucans or β-d-glucans in final products are the various technological processes associated with cereal grains, such as milling, fermentation, extrusion, and germination [46]. The influence of these factors is further discussed below.

## 3. Effect of the Milling Process and Grain Pearling on Cereal β-d-Glucans

Because of the asymmetric distribution of β-d-glucans in cereal grains, primarily favoring the aleurone layer, the milling process significantly affects the yield of β-d-glucans. Harris and Fincher [47] reported that β-d-glucans are concentrated mainly in the subaleuronal cells in the immediate vicinity of the aleurone layer, and their content decreases as they move toward the endosperm proper. A notable exception was observed in barley varieties with a high content of β-d-glucans (waxy, amylose composition of starch). In these varieties, more than 80% of β-d-glucans were primarily located in the endosperm. Additionally, the central endosperm was found to contain less β-d-glucans than its middle layers [48].

The milling process is also influenced by different milling equipment, where targeted sieving and separation can influence the quality of the product, and fractions with high fiber including β-d-glucans, protein, or other substances can be obtained [26]. Intensive mechanical processing, as identified by Wood et al. [49], may result in shearing damage to β-d-glucans, causing changes in their physical properties such as solubility and viscosity. Zheng et al. [48,50] quantified the β-d-glucans content in naked barley after the milling process, with variations ranging between 5%, 8%, and 11% across different sieve passages.

The intensity of grain pearling (also called debranning) before milling significantly influences the composition. When the outer layer of wheat grain was abraded from 3%, representing the pericarp, the total dietary fiber (TDF) content was 62%, with total arabinoxylans (TAX) constituting 26% in this fraction. Increasing the degree of pearling to 12% resulted in a significant decrease in the content of these components, reducing TDF to 22% and TAX to 11% [51]. 

Similar trends of a significant increase in β-d-glucans were possible to identify in the case of oat bran separation. This increase was more than twofold compared to the whole oat grain content [52]. These authors also mapped the effect of different roll milling conditions during bran separation on total β-d-glucans content in oat bran. The study showed that the β-d-glucans content in the bran fractions increased when the break rolls were operated under a dull-to-dull rather than sharp-to-sharp disposition. Only minor changes were observed in the β-d-glucans content for the three separation techniques based on roller and pin milling, as presented in Figure 3A–C. The most significant enrichment effect (1.8- to 2.2-fold increase) in β-d-glucans content was achieved when fine bran was prepared using a roller mill (Figure 3B). The highest β-d-glucans recovery occurred with pin milling techniques, ranging from 75.5% to 78.4% (Figure 3C).

## 4. Impact of Dough Preparation, Fermentation, and Baking on β-d-Glucans

The baking industry supports the trend of incorporating fractions rich in dietary fiber, such as β-d-glucans, into special flours. Nevertheless, it is essential to consider both the workability of the dough and the sensory acceptability of the final bakery product. Simultaneously, the preparation of the dough and its subsequent fermentation and heat treatment may introduce more substantial and diverse changes in fiber content and quality compared to the milling process. Generally, higher fiber content results in higher water absorption, which can cause lower gas retention, lower specific volume of the final product, and lower crumb porosity [26,53,54,55]. On the other hand, the adverse impacts of adding dietary fiber to dough consistency can be mitigated through effective dough preparation techniques, as demonstrated by Jacobs et al. [56] and Kinner et al. [57]. This includes strategies such as pre-hydrating the dietary fiber before incorporating it into the dough and promoting improved water utilization by other biopolymers. Additionally, different baking approaches have been explored as another method for managing these effects, as indicated in research by Andersson et al. [58]. 

The effect of controlled dough fermentation using homo-fermentative lactic bacteria *Lactobacillus plantarum* 22134 on the content, MW, and viscosity of β-d-glucans was studied by Lu et al. [59]. The total β-d-glucans content decreased from 4.89% to 4.23% after 12 h of fermentation. On the other hand, soluble β-d-glucans concentration increased from 1.89% to 2.18% and then decreased to 1.97% after 8 h of fermentation, which is in a high positive correlation with viscosity. After 4 h of fermentation, a decrease in β-d-glucans with a MW exceeding 10^5^ g.mol^−1^ was noted, and a significant reduction in this fraction was further observed after 12 h of fermentation. These changes are likely associated with parallel processes involving the activity of naturally occurring endogenous β-glucanases in cereal flour, responsible for degrading insoluble forms of β-d-glucans, and the activity of lactic acid bacteria, which consume the soluble form of β-d-glucans [60,61,62]. 

Baker’s yeast (*Saccharomyces cerevisiae*) is also employed in the baking industry for fermentation. Lu et al. [59] noted that sourdough bread exhibits better potential than yeast-fermented bread in terms of preserving the MW and viscosity of β-d-glucans during the baking process. Andersson et al. [63] reported a contrasting effect of yeast. The degradation of β-d-glucans during mixing, fermentation, and proofing is attributed to endogenous β-glucanases in cereal flour, but it is not influenced by the presence of yeast. Considering the reported degradation of β-d-glucans with higher MW, especially during longer fermentation cycles [59], it is advisable to incorporate this size fraction in a higher proportion during flour fortification. This approach allows simultaneous utilization of the extended fermentation period, facilitating the redistribution of water from non-starch polysaccharides back into the gluten structure and thereby improving the specific volume of the bread [26,56].

There is some controversy regarding changes in the MW of β-d-glucans after fermentation. Several studies suggest that fermentation may lead to a decrease in β-d-glucan content, as observed in barley grains [64] and oats [65]. Some authors [58,66,67] further showed a decrease in the MW of β-d-glucans and their content in barley and rye dough with a fermentation time of 0–60 min and in a sourdough from barley whole meal flour and oat bran leaven under fermentation conditions of 30 °C for 20 h. Andersson et al. [58] and Rieder et al. [67] further demonstrated a reduction in the MW and content of β-d-glucans in barley and rye dough during fermentation times ranging from 0 to 60 min. Similar observations were made in a sourdough prepared from barley whole meal flour and oat bran leaven, fermented under conditions of 30 °C for 20 h. On the contrary, Comino et al. [68] reported no significant changes in the content of arabinoxylans and β-d-glucans, as well as in the arabinose/xylose ratio for arabinoxylans and the cellotriosyl/cellotetraosyl units for β-d-glucans during dough preparation and fermentation. The results from [69,70] confirmed that fermentation at 30 °C for 20 h with added lactobacilli led to a reduction in dietary fiber and β-d-glucans content. However, the MW was not affected. Lambo et al. [70] also reported that the reduction in total β-d-glucans content is associated with the strain of microorganisms used in the fermentation.

The information regarding the contribution of β-d-glucans to the technological quality of leavened bakery products is also not entirely consistent. For instance, the inclusion of barley flour or β-d-glucans-rich fractions in baker’s strong wheat flour led to a reduction in specific bread volume [53,55]. On the other hand, the positive impact of β-d-glucans on increasing loaf volume through improved gas retention was confirmed by the authors [71,72].

Thus, the method and optimization of fermentation time can be crucial for the composition and properties of β-d-glucans, as well as for the actual technological quality of the final bakery products. A potential strategy to stably preserve the original structure of β-d-glucans is to incorporate these components after the dough fermentation [63]. 

The impact of baking on whole meal yeast-raised rye bread was investigated, revealing no significant changes in the contents of arabinoxylans, β-d-glucans, and arabinogalactans [68]. The authors noted only a slight increase in the solubility of arabinoxylans and β-d-glucans. Additionally, Maina et al. [61] affirmed that, in general, baking has no discernible effect on the molecular structure (cellotriosyl/cellotetraosyl ratio) of β-d-glucans or their overall amounts. Andersson et al. [58] reported a slight decrease in the MW β-d-glucans during baking. However, Tiwari et al. [73] and Rakha et al. [74] observed a rapid decrease in the HMW of β-d-glucans, coupled with an increase in the medium (MMW) and low-molecular-weight (LMW) forms, attributed to baking time and temperature. 

Another method of processing cereals with higher β-d-glucans content, such as oats and barley, involves microbial fermentation to create new functional foods (e.g., porridges, yogurts, and beverages) that add value to consumer health. Wet enzyme-based fractionation processes and fermentation can enhance the levels, digestibility, and bioavailability of health-related components in the human diet. Additionally, these processes contribute to improving the structural features and sensory properties of the food while concurrently reducing the concentration of unwanted components [65,75]. An updated review of β-d-glucans changes in oats is provided by Djorgbenoo et al. [76]. The literature provides conflicting evidence on the effect of fermentation on the content of β-d-glucans, a key active constituent of oats. Opinions vary from data showing no significant effect on β-d-glucans content to observed reductions of 17–50%. In agreement with Djorgbenoo et al. [76], further research is needed to understand the factors affecting β-d-glucans content during fermentation and to elucidate its implications for the nutritional properties of fermented oats. 

## 5. The Influence of Germination on β-d-Glucans in Cereals

Sprouting is a traditional processing method that has been used for centuries to improve the nutritional value of cereals and legumes. Sprouting is also alternatively known as malting because this process has been used for centuries to produce malts with distinctive functionality and flavors for the brewing and distilling industry. The growing interest in this method in recent years is due to a high demand for more natural and healthy foods [77]. Sprouting can induce changes in the nutritional composition through enzymatic hydrolysis of existing compounds and de novo synthesis of new compounds [78]. It is namely associated with increased iron, zinc, and calcium bioaccessibility, reduction in phytate, and better digestibility of big molecules such as proteins or polysaccharides. 

Germination is used in the food industry also to improve the texture, flavor, and sensory characteristics of cereals [79]; so recently, the putative health-promoting effects of sprouted products have increased their application [80]. Sprouted cereal grains can be added to breakfast foods, yogurts, salads, soups, and smoothies or milled into flour for bakery products such as bread, noodles, pasta, biscuits, and tortillas [81]. The increased digestibility and lower viscosity of sprouted products have also been utilized in the preparation of weaning and geriatric foods because of acceptable digestibility [82]. However, the increase in enzymatic activity during sprouting can also negatively impact the baking properties and consumer acceptability of bakery products produced from sprouted flour [83]. Sprouted flour has been linked with decreased dough development, stability, water absorption, and a sticky and wet crumb [84]. The quality of malt is also closely associated with germination and the effective degradation of β-d-glucans. Elevated levels of β-d-glucans in malt are undesirable, despite their nutritional benefits. Thus, the different phases of β-d-glucans degradation, including the action of exo- and endo β-glucanase to achieve the desired β-d-glucans content in barley malt (100 to 300 mg.L^−1^), have been studied in detail in the brewing industry and are constantly evolving [85].

Dietary fiber is an important component of cereals of which β-d-glucans and arabinoxylans belong to soluble dietary fiber and cellulose and lignans to insoluble dietary fiber. Sprouting solubilizes fiber due to the activation of endogenous enzymes, which led to a decrease in insoluble fiber and an increase in soluble fiber from 1- to 2-fold in sprouted grains of wheat and barley, although total dietary fiber decreased [86]. 

Oats contain high levels of β-d-glucans that decrease in sprouted grains due to the activation of β-glucanase [65,79,87,88], but the decrease in β-d-glucans is initially very slow [79]. In the case of the high-beta-D-glucans barley variety, the decrease in the content after 24 h of germination was also gradual (about a 13% relative decrease). However, after 96 h of germination, the decrease was already very significant, from the initial 10.5% to 4.31% [89]. Therefore, short sprouting times are recommended to retain soluble fiber and β-d-glucans in oats [87]. 

## 6. Effect of Extrusion

Extrusion cooking is a continuous process wherein food materials are subjected to high pressure, high temperature, and intense mechanical shear in a short time. This method is commonly employed to produce expanded cereal products. Various chemical reactions and the disruption of cell wall structures are noted including changes in the properties of major grain components and major polymers, such as starches, proteins, and fiber [61]. 

Extrusion causes irreversible changes in the flour components, especially the flour polymers, i.e., starch, protein, and fiber, by breaking the polymer chains. The disruption of polymer chains leads to a decrease in MW, the formation of starch–lipid, protein–lipid, and protein–protein complexes, and an increase in the solubility of fiber and starch [90]. Limited information is available on the direct assessment of changes in the properties of β-d-glucans after extrusion compared to, for example, wheat arabinoxylans [61]. Sharma and Gurjal [91] investigated the impact of extrusion cooking on barley, revealing that the total β-glucan content remained unaffected. However, a notable increase in soluble β-d-glucans was observed, with the highest levels noted in extrudates produced under high-temperature, low-moisture conditions. The ratio of soluble to insoluble β-d-glucans varied from 0.7 to 1.5 in the control barley, but after extrusion cooking, the ratio was changed from 1.2 to 3.1. The β-d-glucans extractability increased by up to 8% as well. It was also observed that extrusion caused changes in the molecular mass and solubility of the β-d-glucans. Moreover, microscopic examination revealed that harsher extrusion conditions caused depolymerization, disintegration of cell walls, and affected the distribution of β-d-glucans in the cereal. The reduction in MW of β-d-glucans had an impact on the hardness and density of the extruded cereals and increased gut viscosity [92,93]. 

In the context of dietary fiber, of which β-d-glucans are a vital component, several changes occur during extrusion. During extrusion, thermomechanical changes, depolymerization, changes in solubility, and transglycosidation processes have the effect of reducing the insoluble dietary fiber and TDF content; on the other hand, they increase the content of soluble dietary fiber [68,90,94,95,96]. As a result of extrusion, a higher loss in insoluble dietary fiber is observed compared to the increase in soluble dietary fiber. This is explained by the conversion of cellulose to lower molar mass soluble dietary fiber fractions that could be further degraded into low MW components and sugar fractions that cannot be determined as dietary fiber [94]. During extrusion, TDF content can be increased by the formation of resistant starch or indigestible Maillard reaction products. Increasing the extruder screw speed increased the soluble dietary fiber content. The soluble dietary fiber content of the product and the MW can be influenced by setting the temperature and the initial addition of water [90]. The MW of soluble dietary fiber plays an important role in the expansion rate [90,97]. The results of the increase in water-soluble fiber components after extrusion may be related to the increase in water-soluble β-d-glucans and arabinoxylans found by Andersson et al. [63], Comino et al. [68], Sharma and Gujral [91], and Gujral et al. [98] in barley, wheat, and rye. In contrast, no changes were observed by these authors in the ratio of basic building units. 

## 7. The Range of Applications of β-d-Glucans in Food Products 

Cereal β-d-glucans, mainly found in barley and oats, possess technological advantages and multiple health benefits. However, changes in their physicochemical properties and health-promoting effects can occur during food processing and storage, representing a major disadvantage [99]. To maximize the health benefits of β-d-glucans for a wide range of consumers, it is essential to consider the most salient aspects of buying and consuming food products. This includes ensuring a high textural and sensory quality of the product [100]. To maximize the potential of cereal β-d-glucans, it is essential to extend their applications beyond bread and cereal beverages. Taking this action will allow the realization of the complete range of benefits linked to this polysaccharide [99].

β-d-Glucans have potential applications in medicine, pharmacy, veterinary, cosmetic, and chemical industries and in food and feed production. The biological activity and effects of β-d-glucans depend significantly on their source, the isolation method used, and their specific physicochemical properties, such as structure, viscosity, and MW [101]. The consequential technological operations in the production of food and food supplements also have a significant influence. β-d-Glucans are valuable functional ingredients because they can influence the quality, structure, rheological properties, and stability of various food systems, and they have various physical properties; they are capable of thickening, stabilizing, emulsifying, and gelling [6,102]. The addition of β-d-glucans to foods has been shown to improve the physical properties of food products. β-d-Glucans affect the consistency of the product and can be used as fillers, stabilizers, thickeners, and fat replacements in many dairy and bakery products, improving sensory quality or extending the shelf life of the product and slowing down the aging process. In terms of human nutrition, they reduce energy and increase the nutritional value of foods [103]. The naturally occurring β-d-glucans in cereals present an attractive opportunity for manufacturers to design foods with added health value and positive labeling for consumers [6]. 

Many ways of using cereal β-d-glucans in the production of specific foods are available in the literature (Table 1). Due to their emulsifying and thickening properties, LMW β-d-glucans [104] are commonly used successfully in liquid food matrices, such as beverages, sauces, and soups [13,105,106]. A drink was developed (1 g of HMW oat β-d-glucans in 250 mL of water) with a positive effect on reducing the level of LDL cholesterol in blood serum [13]. LMW β-d-glucans have less impact on the product’s final viscosity [103]. The addition of cereal β-d-glucans to chocolate-flavored milk has been demonstrated to be successful in improving the texture, mouthfeel, and taste of the chocolate, as well as increasing the viscosity of the product. At a concentration of 3%, the cereal β-d-glucans were able to enhance the overall quality of the beverage [107]. 

Recently, there has been growing interest in using oats as a raw material in brewing, driven by research confirming their safe consumption in appropriate quantities for individuals with coeliac disease and offering unique sensory properties. It was tested to produce beer from oat malt, barley malt, and a combination of oat and barley malts with varying amounts of unmalted oats (22.5 and 45%). Malted oats in the wort resulted in lower extract content, higher protein content, and lower free amino nitrogen compared to barley wort. Oat wort exhibited increased viscosity and haze because of the presence of β-d-glucans [108]. The next study revealed that incorporating a 10% share of oat malt had no adverse effects on the brewing process or final product quality compared to the recipe without oat malt. However, higher oat malt proportions resulted in a prolonged lautering time (up to approximately 120 min), reduced wort quantity (up to about 30%), lower extract content (up to about 35%), leading to decreased alcohol concentration, and intensified beer color [109].

Angelov et al. [110] found that the production of a cereal-based fermented beverage did not result in any changes in β-d-glucans content during fermentation and storage. This suggests that the β-d-glucans present in the beverage were unaffected by the fermentation and storage process. 

For solid food matrices, the addition of β-d-glucans is mainly investigated for cereal-based products. The addition of β-d-glucans to bakery products is known to lead to an increase in dough viscosity and crumb strength, as well as improved elasticity and coloration [111]. Bread with high β-d-glucans content had higher loaf volume and lower glycemic index [112]. Bread with added oat β-d-glucans has a lower specific volume and porosity, darker color, higher hardness, and lower elasticity and consistency than the control white bread. However, these negative effects can be effectively counteracted by optimizing the water content of the products [113]. The baking properties most affected by β-d-glucans were volume and color. Oat β-d-glucans softened the crust of gluten-free bread but had the opposite effect on wheat bread [114]. High-glucan barley flour has been used in the production of pasta at a rate of 30% [115] and in the production of couscous [116]. 

The fortification of yellow alkaline noodles with barley β-d-glucans has been shown to reduce their sensory quality [117]. However, when a new sponge cake formulation was tested with a 4.4% addition of β-d-glucans, sponge cakes had excellent consumer acceptability [118]. An even higher addition of 5.2% β-d-glucans in a cereal bar led to an improvement in taste and overall appearance ratings [119]. According to Zbikowska et al. [120], an increase in β-d-glucans content, accompanied by a 5% reduction in fat, regardless of the type (microbial, yeast, or oat), led to a deterioration in the quality of biscuits and induced changes during storage. Additionally, barley β-d-glucans have been used as a fat substitute in yogurt and have been able to improve the texture and sensory quality with just a 0.5% addition [121]. 

The use of β-d-glucans in cheese production is interesting. β-d-glucans can structure the mixtures during the fermentation of the lactic curd, actively bind the whey, and result in higher yields of cheese produced [122]. Cereal β-d-glucans have the ability to replace the texture and taste of milk fat in products but impair the sensory properties of protein-containing products [123]. β-d-Glucans have been found to reduce the acidity of cheese and increase the yield of products by preventing whey separation and contributing to texture formation. However, grain β-d-glucans can significantly impair the sensory properties of cheese, be a source of flavor, and change the color of the dairy product [124,125]. Due to the water-retention ability of β-d-glucans, they have a significant ability to reduce losses, increase food product yields, and reduce food process times [126].

**Table 1 foods-13-00500-t001:** Examples of β-d-glucans applications in specific food processing.

Application	Addition	Source	Aim	Product	References
Cereal processing	2.5 and 5%	Barley	Fortification with soluble fiber, increasing loaf volume	Bread	[102]
Cereal processing	0.2, 0.6, 1.0, and 1.4%	Barley	Fortification with soluble fiber, resistance to deformation	Bread	[71]
Cereal processing	10, 12, and 14% soluble oat fiber (70% β-d-glucans)	Oat	Replacing of flour in wheat bread	Bread	[113]
Cereal processing	2.6 and 5.6%	Oat	Addition of functional ingredients	Bread	[114]
Cereal processing	30% high-beta-D-glucan barley flour	Barley	Fortification–health claim linking the consumption of barley beta-D-glucan	Pasta	[115]
Cereal processing	10%	Oat	Fortification	Yellow alkaline noodles	[117]
Cereal processing	20 and 30% high-beta-D-glucan barley flour	Barley	Fortification-functional couscous	Couscous	[116]
Cereal processing	100% barley flour (3.4–4.4% β-d-glucans)	Barley	Fortification, replacing of wheat flour	Sponge cake	[118]
Cereal processing	5.2%	Barley	Fortification with dietary fiber, reduction in energy	Biscuit bar	[119]
Milk processing	0.5%	Barley	Fat replacement	Yogurt	[121]
Milk processing	0.2–0.8%	Yeast	Thickening	Yogurt	[127]
Milk beverage production	3%	Oat	Stabilizing	Chocolate-flavored milk	[107]
Cheese production	0.7 and 1.4%	Oat	Fat replacement	White-brined cheese	[125]
Cheese production	Fat replacement 3.47% and 6.84%	Yeast	Fat replacement	Cheddar cheese	[128]
Cheese production	0.5%	Barley	Fat replacement	Dahi cheese	[129]
Cheese production	5%	Barley	Fat replacement	Labneh	[130]
Cheese production	0.2%	Barley	Fat replacement	Mozzarella	[131]
Milk processing	0.5%	Cereal (not specified)	Fat replacement	Cottage cheese	[132]

## 8. Potential of Cereal β-d-Glucans in Nanotechnology

Nanotechnology is a newly emerging technique, which involves the characterization, fabrication, and manipulation of structures, devices, or materials that have at least one dimension having 1–100 nm in length. Nanomaterials are further clustered into three classes, namely, nanoparticles, nanofibers, and nanoplates. Nanomaterials have unique properties, unlike their macroscale counterparts due to the high surface-to-volume ratio and other novel physiochemical properties like color, solubility, strength, diffusivity, toxicity, magnetic, optical, and thermodynamic characteristics. Nanotechnology also provides a wide range of opportunities for the development of new products and applications, for example, in the therapeutic and nutritional fields. Some of the main targets in the use of nanoparticles in food industries can be to improve organoleptic characteristics, increase absorption and intentioned delivery of nutrients and bioactive compounds, and stabilization of active ingredients such as nutraceuticals in food structures [133,134]. 

Medical applications permanently seek tools against various diseases. Based on albumin, peptides, and polysaccharides, therapeutic compounds are developed that help to deliver the active substance to a defined site in the body, amplify its effects, and directly activate components of the immune system or as imaging agents. Nanotechnological processes that purposely reduce the particle size lead to greater adhesion of the surface of the nanocarrier and thus result in higher and more controllable bioavailability of the incorporated active agent [135,136,137]. Some of these therapeutic principles can be well applied to the production of functional foods, where nanomaterials can act as nutraceuticals, nanocapsules, nanoemulsions, or as protective ingredients in food packaging [134].

### 8.1. Characteristics of β-Glucans from Different Sources for Use in Nanotechnology

Therapeutic and nutritional utilization of nanomaterials demands their renewability, nontoxicity, biocompatibility, biodegradability, and potential biological ability. These properties are absolutely fulfilled by β-glucans as a heterogeneous natural polymer of non-starch polysaccharides, which are more likely to be compatible with the biosystems inside the human body. Furthermore, β-glucans have attracted much attention as a wall material due to their helix honeycomb structure, accommodating bioactive ingredients easily [135,136].

β-Glucans from different sources exhibit considerable variability in structure, encompassing differences in MW, degree of branching, and types of linkages. Apart from the cereal β-d-glucans discussed in the preceding section, other developed forms include branched β-(1,3; 1,6)-glucans found in brown algae (known as laminarin), baker’s yeast, and fungi such as *Lentinus edodes* (known as lentinan) and *Schizophyllum commune* (known as schizophyllan). Various other structures are also recognized, such as cyclic β-(1,3; 1,6)-glucans and hyperbranched β-(1,3; 1,4; 1,6)-glucans present in bacteria of the family *Rhizobiaceae* and sclerotia of the fungus *Pleurotus* tuber-regium, respectively. At the same time, it is also confirmed that the diverse chemical structures of β-glucans can be suitably exploited in specific nanotechnology applications as described by Li and Cheung [2]. Results published by Vetvicka and Vetvickova [137] indicate that the quality of isolation (purification), rather than the origin, is the major influence on the levels and ranges of the biological activity of β-glucans from different sources. Specific differences in immunological activity, blood sugar, and cholesterol-lowering effects of isolated β-glucans from various sources (oats, yeast, and fungi) and different levels of purification were confirmed by the study. 

### 8.2. Biological Activities of β-Glucans Applied in Nanotechnology 

In their recent work, Yang and Cheung [17] highlighted over 200 therapeutic nanoapplications primarily utilizing fungal β-glucans. These β-glucans, derived from fungi and yeast, have demonstrated remarkable anticancer and immunomodulatory properties among β-glucans from various sources. Moreover, in the past decade, researchers have unveiled some of the mechanisms underlying the anti-inflammatory effects of β-glucans, particularly through the stimulation of immune memory cells. 

Hwang et al. [136] further highlight the limitations of conventionally prepared β-glucans, which are polysaccharides characterized by their large MW. These properties have traditionally hindered their utility as drugs, proteins, and supports for DNA transfer. The challenges arise from their high viscosity, potential instability, low solubility, short half-life, and limited cell permeability. Consequently, they result in exceedingly low blood concentrations, rendering them incapable of triggering an immunological response.

However, these unfavorable characteristics can be effectively mitigated through the appropriate preparation of nanoparticles, micelles, and nanocapsules spanning a size range of 10 to 500 nm. Such tailored structures offer the potential to enhance therapeutic efficacy by entrapping or conjugating therapeutic drugs and forming stabilized drug substances. For β-glucans, commonly employed methods involve electrical spinning or treatment with acid (TFA) or DMSO, which serve to convert the macromolecules into an LMW and a single-fiber state [17,137]. Moreover, Ashraf et al. [138] reference other studies that have delved into the production of β-glucan nanoparticles using various methods, including supercritical fluid, emulsion, and solvent techniques.

Of all the β-glucans from various sources, those originating from mushrooms and yeast have been shown to possess remarkable antitumor and immune-modulating properties [17]. Within this context, earlier investigations have conjectured that the degree of branching plays a pivotal role in the biological activity of β-glucans. Specifically, unbranched β-glucans exhibit negligible activity, whereas their branched counterparts demonstrate some level of bioactivity [139]. Furthermore, Zeković et al. [103] have additionally suggested that β-glucans with an exceedingly LMW fail to exhibit significant biological activity. 

The comparison of bioactivity including other properties of β-glucans from different sources taken from Peltzer et al. [140] is summarized in Table 2. According to this comparison, cereal β-d-glucans do not mainly show an immunomodulatory function or have an anti-inflammatory effect. On the contrary, together with mushroom and yeast β-glucans, they show an effect on lowering blood sugar levels, and only cereal β-d-glucans are attributed to lowering blood cholesterol levels, which is also mentioned by Lante et al. [141].

### 8.3. Modifications of the β-Glucans Molecules for Nanotechnology

The advantage of branched fungal β-glucans primarily stems from their higher number of hydroxyl groups, serving as key reactive sites for structural modifications. Yang and Cheung [17] highlighted various modification techniques such as sulfation, carboxymethylation, acetylation, hydroxyethylation, hydroxypropylation, methylation, amination, and esterification that can be applied to these hydroxyl groups. These characteristics empower branched fungal β-glucans to serve as a foundation for constructing β-glucan-based nanoparticle systems. These systems not only exhibit enhanced physiological properties but also possess a remarkable capacity for loading immune-modulating agents, thereby broadening their therapeutic potential.

The realm of nanotechnology research exploring the comparative effectiveness of β-glucans from diverse sources in identical applications remains notably limited. One illuminating instance of such a comparative analysis is exemplified by the study conducted by Ashraf et al. [138]. In this study, the impact of β-glucan nanoparticles derived from oats, barley, and yeast was systematically assessed across a spectrum of parameters encompassing antioxidant, antidiabetic, antihypertensive, antiproliferative, and antimicrobial activities. A significantly conclusive effect of the nanoparticles on the increase in the above-observed biological activities was confirmed. Significantly lower but detectable biological effect was also observed among the β-glucan sources. While oat β-d-glucan nanoparticles exhibited heightened anticarcinogenic activity when compared to their yeast or barley counterparts, they concurrently displayed the lowest amylase inhibition and antidiabetic activity. Conversely, yeast-derived β-glucan demonstrated the highest antimicrobial activity, albeit with the lowest antihypertensive efficacy in this comprehensive comparative analysis. 

### 8.4. β-Glucan Nanotechnology in the Food Industry 

A recent interesting study comparing the detoxifying properties of oat and yeast β-glucan applied to milk with the presence of aflatoxins (AFM1) was performed by Pavlek et al. [142]. The maximum and mutually comparable final AFM1 decontamination (0.169 μg.kg^−1^ for oat and 0.178 μg.kg^−1^ for yeast β-glucan) was achieved in the first hours of binding of both β-glucans but in different concentrations (0.01% for oat compared 0.005% for yeast β-glucan). This is probably due to the different molecular structure of β-d-glucans from oats compared to β-glucans isolated from yeast.

The use of cereal β-d-glucans in the food industry with various functional properties such as thickening, emulsifying, stabilizing, and gelling is quite common. On the other hand, their processing by nanotechnology in both food and biomedical applications still has considerable reserves compared to fungal β-glucans. The production of nanoparticles obtained through super fine grinding was investigated for oat β-d-glucans by Liu et al. [143]. The results showed that superfine grinding treatment could decrease OBP’s (oat bran polysaccharides) MW and apparent viscosity and enhance OBP’s solubility and antioxidant activities, suggesting that superfine grinding technology had great potential to be applied in functional polysaccharide food. 

Another example is barley β-d-glucans processed into nanoparticles for encapsulation of α-tocopherol and its release in vitro during transit in the gastrointestinal tract. The enhanced release of α-tocopherol from Glu-N (β-d-glucan nanoparticle) may be related to its reduced particle size, which makes the encapsulated compounds released more readily and in a shorter time than from larger particles [144].

Salmerón et al. [131] emphasize that the food industry is actively pursuing novel synthesis methods to enhance the integration of medically advantageous components into various food matrices. These components encompass, e.g., zinc oxide particles (known for their antidiabetic effects), gold nanoparticles (recognized for their anticancer properties), as well as nanoinorganic metal oxides and silver nanoparticles (acknowledged for their antibacterial properties). In this context, nanotechnologically modified cereal β-d-glucans could also potentially assume the role of a carrier or a key component in nanocapsule formulations, concurrently bestowing probiotic activity, thus expanding its potential applications in the food industry. 

A noteworthy illustration of oat β-d-glucans utilization, characterized by an average particle size of 465 nm, as a biotherapeutic agent in zebrafish larvae challenged by the pathogenic bacterium *Edwardsiella tarda*, is provided in the study conducted by Udayangani et al. [145]. Generally, superfine grinding is a new technology, which is a useful tool for making superfine powder normally with a particle size of less than 10 μm and good surface properties. Superfine oat bran was prepared according to the following procedure. Dried oat bran was milled in a common pulverizer and sieved through a 40-mesh screen. Then, the oat bran was further ball-milled in a CJM-SY-B type high-energy nano-mill for 8 h. The particle size of the superfine ground oat bran powder ranged from 200 to 620 nm, with an average particle size of 397.6 nm. This research not only underscores the effectiveness of cereal β-d-glucans in immunomodulation but also serves as evidence for their ability to combat certain harmful microorganisms.

In conclusion, this chapter suggests that the nanotechnological processing of cereal β-d-glucans could broaden their applications in both immunomodulation and harnessing their anticarcinogenic properties. At the same time, their relatively higher abundance in the waste bran from the milling of oats and barley, coupled with ongoing process optimization, makes cereal β-d-glucans readily available for more efficient isolation and subsequent nanotechnological processing in the food, pharmaceutical, and cosmetic industries. 

## 9. Conclusions

Mechanical approaches, especially optimized debranning, emerged as highly effective for obtaining substrates with naturally concentrated β-d-glucans. While opinions on the impact of fermentation processes on β-d-glucans structure and properties vary, further research in this direction is crucial. Germination, notably, leads to a gradual breakdown of β-d-glucans due to β-glucanase activation, making shorter germination periods more promising for preserving higher β-d-glucans levels. 

Among thermal processes, extrusion exhibited the most significant influence on β-d-glucans content and properties, leading to depolymerization and increased solubility. The utilization of cereal β-d-glucans extends beyond traditional applications in cereal products, showing potential in medicine, pharmacy, veterinary, cosmetics, and chemical industries. 

In the food industry, β-d-glucans enrichment, combined with recipe optimization, significantly improves various physicosensory properties. For instance, in bakery products, β-d-glucans enhance dough viscosity, crumb firmness, elasticity, and color. Incorporating β-d-glucans into various foods, such as flavored milk, biscuits, cereal bars, and yogurts, successfully enhances texture, taste, and overall quality. However, some studies also show that a higher addition of this polysaccharide in the food industry can negatively affect some processes and properties of bakery products and fermented beverages. The more frequent inclusion of β-d-glucans in various product categories will depend on economically effective β-d-glucan separation, ensuring it does not significantly raise the final product’s cost.

Nanotechnology emerges as a game-changer, opening new possibilities for β-glucans applications, especially in medicine and the food industry. While fungal or yeast β-glucans dominate current nanotechnological applications, cereal β-d-glucans present promising objects for nanotechnological processing. Despite the limited studies comparing cereal β-d-glucans with counterparts from other sources in nanotechnological applications, it is evident that nanotechnological approaches significantly expand the application potential of cereal β-d-glucans, particularly in areas like immunomodulation and anticancer effects.

β-d-Glucans from cereal sources are relatively easily available, relatively stable, and widely usable in health and the food industry. Their intake into the human body in the approved amount is not characterized by any negative effects. These safety advantages open new possibilities for research and use of this plant metabolite in various industries. Looking ahead, waste bran material from oats and barley stands out as a prospective source for isolating and nanotechnologically processing cereal β-d-glucans, potentially accelerating their application in nanotechnology, food, pharmaceuticals, and cosmetics.

## Figures and Tables

**Figure 1 foods-13-00500-f001:**
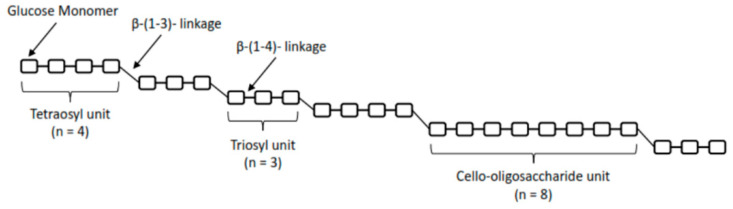
Structure of cereal β-d-glucans with β-(1-3)- and β-(1-4)- linkages and the “staircase”-like structure, adapted from Henrion et al. [6].

**Figure 2 foods-13-00500-f002:**
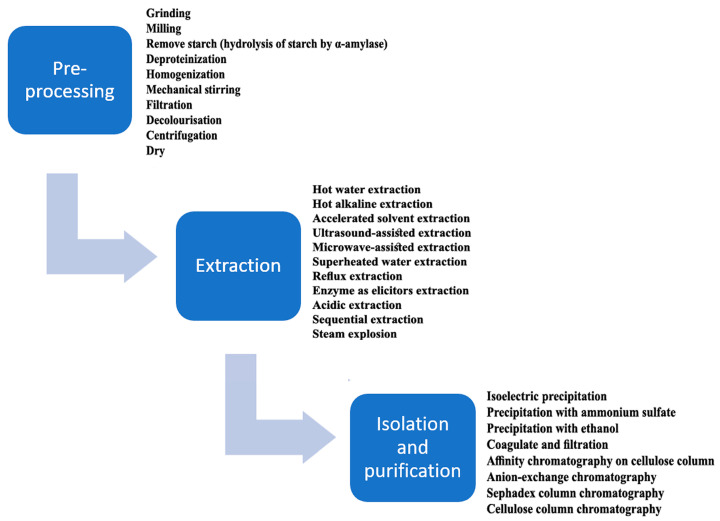
The basic steps in the production of β-d-glucans from cereals adapted from Zhu et al. [19].

**Figure 3 foods-13-00500-f003:**
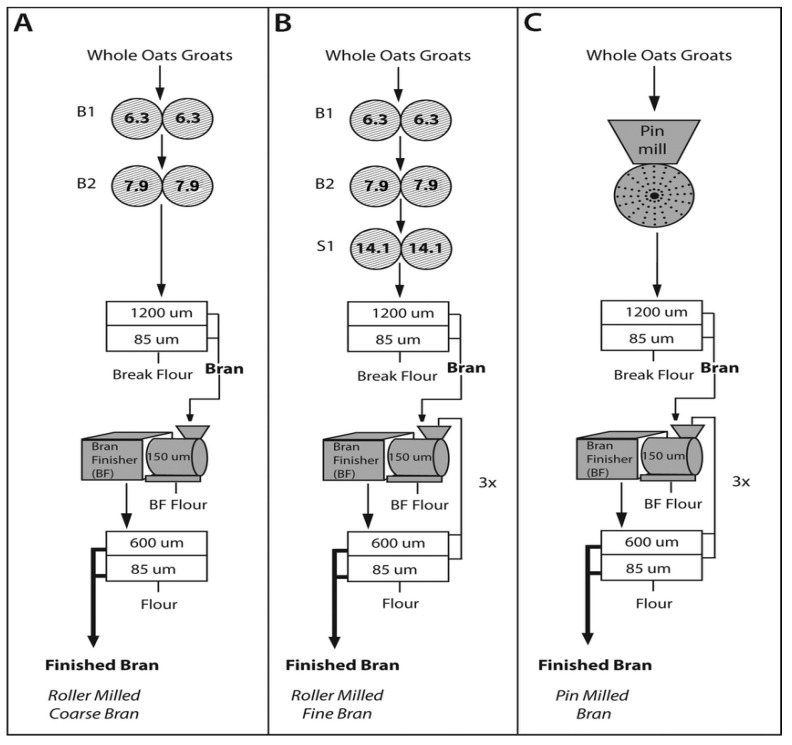
(**A**–**C**) Three ways of oat bran preparation using roller and pin milling techniques [52]. Three bran types were generated through roller and pin milling methods. Initially, groats were coarsely ground using corrugated rolls, followed by further size reduction with finer corrugations. Oat bran was formed from fractions retained on 600 μm and 85 μm sieves, while the fine fraction became break flour. Roller-milled coarse bran (**A**) combined coarse fractions from the second break system, roller-milled fine bran (**B**) involved narrowing gap sizes and using additional sizing rolls, and pin-milled bran (**C**) was produced through pin milling and subjected to sieving and bran finishing processes similar to roller-milled bran [52].

**Table 2 foods-13-00500-t002:** β-Glucan properties from different sources (adapted according to Peltzer et al. [140]).

β-Glucan Properties	β-Glucan Sources				
	Oat	Barley	Mushrooms	Brewer’s Yeast	Baker’s Yeast
Structure	β-1,3;1,4 glucans	β-1,3;1,4 glucans	β-1,3;1,6 glucans	β-1,3;1,6 glucans	β-1,3;1,6 glucans
Reduce serum cholesterol levels	X				
Attenuate blood glucose level	X	X			X
Improve/stimulate immune function			X	X	X
Promotes healthy inflammatory response			X		X
Topical application/skin treatment/wound healing					X
Mycotoxin adsorption	X	X		X	X
Fat replacer in food formulations				X	X
Food thickener, emulsion stabilizer	X	X			X
Film-forming biopolymer	X	X	X	X	X

## Data Availability

The data presented in this study are available on request from the corresponding author.

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
