# Peer review of "Cereal β-d-Glucans in Food Processing Applications and Nanotechnology Research"

_foods, 2024, doi:10.3390/foods13030500_

Round 1
Reviewer 1 Report
Comments and Suggestions for Authors
This review outlines the extraction methods of cereal beta-glucans, and the effects of milling process, dough fermentation and baking, sprouting treatment and extrusion on their function and properties. The food application range and nanotechnology of cereal beta-glucans were also summed up. The summary work is worth doing. Suggestions,
1) Among 140 refs, only 57 refs are from the year of 2015. During 2007 to 2010, there were many articles including reviewing paper on cereal beta-glucans in Cereal Chemistry and Journal of Cereal Science. The refs should be chosen in the past 10-15 years.
2) Many places are irregularly expressed, like Line 100, to enhance the concentration of naturally occurring β-D-glucans, [22] employed ultrafine grinding and electrostatic separation techniques;
Line 182, by [48];Line 295, by [75];Line 298, with [75]; Line 381, by [62,67,90,97]; Line 524, taken from [135]; Line 563, by [138]; Line 584, by [140], Line 594, according to [135].
3) In table 2, the glucan should be expressed on beta-1,3;1,4, not beta-1,3/1,4, the latter forms is not seen in plant cell wall references.
4) Line 57, LDL; Line 512, DMSO should be noted.
5) In figure 2, words are too small to read.
6) In table 1, the function or aim of beta-glucans addition should be given.
7) Line 302-340, the barley malt and oat beverage should be more detailedly evaluated. Oat beverage expands in recent years.
8) Line 602, please check “debrening”.
9) To the nanotechnology process concept on cereal beta-glucans, the authors refer to superfine grinding technology. How to realize nanotechnology process? The authors should sum up the pathways in line 588 to 593.
Author Response
Dear reviewer, the authors team would like to thank you very much for taking the time to read the manuscript and to give useful suggestions. We tried our best to accept all your suggestions and according them to improve the quality of the manuscript.

Reviewer 2 Report
Comments and Suggestions for Authors
The manuscript (“Cereal β-D-glucans in advanced technologies and nanotechnology application”), touches on a very current issue of cereal β-D-glucans found in oats and barley. The problem undertaken at work is interesting, however, in the manuscript there are some places that must be revised. Title clearly describes what the manuscript is about. Abstract are not adequately describes the work – should be corected. Most of the data of literature are properly analyzed and interpreted, but there are lack of important information’s.
The conclusion part, should be corrected. Most of the literature is from the last 10 years. Cited references not always correct. Consequently, I think minor revision study is necessary.
Reviewer's suggestions below:
L.37: “- see Fig. 1 adapted according to [6]” should be replaced with: “(Fig. 1)”
L.42: “adapted from [6].” should be replaced with: “adapted from Henrion et al. [6].”
L.74-75: „The schematic … is presented in Fig. 2. (adapted according to [18]).” should be replaced with: „The schematic … is presented in Fig. 2.”
L.167: „The basic steps in the production of β-D-glucans from cereals adapted from [18]” should be replaced with: „The basic steps in the production of β-D-glucans from cereals adapted from Zhu et al. [18]”
L.201: (Figure 3B) and L.202: (Fig. 3C) - what is correct?
L. “In agreement with [75], further…” should be replaced with “In agreement with Djorgbenoo et al. [75], further…”
Regarding „The range of applications of β-D-glucans in food products”
This part should be supplemented with new information regarding the use of beta glucan in food products. Cited literature could be supplemented by the new positions, for example: doi.org/10.3390/molecules27041393; doi:10.5507/bp.2007.038;
Regarding Conclusion
Summary, should include only conclusions resulting from the work results.
L.597-601 “This review extensively covered ….. aspects of enzymatic extraction as particularly promising.” - These information should be removed, because are not the conclusions of the presented analysis.
Authors write: “In the food industry, β-D-glucans enrichment, combined with recipe optimization, significantly improves various physico-sensory properties. For instance, in bakery products, β-D-glucans enhance dough viscosity, crumb firmness, elasticity, and color.” Meanwhile, some important scientific works have proven, that a larger addition of beta glucan worsens the properties of bakery products.
Regarding References
References needs to be adapted to the editorial requirements of the “Foods” journal.
Author Response
Dear reviewer, the authors team would like to thank you very much for your efford, time, and energy to read carefully the manuscript and to give us usefull remarks. We tried our best to accept all of the and to correct the text according to your suggestions.

Reviewer 3 Report
Comments and Suggestions for Authors
The manuscript presents a comprehensive overview of cereal β-D-glucans, from chemical structure to applications and health benefits. It is generally well-written. However, the title is not in line with the manuscript content. Also, the sections are too long. Authors should consider organizing them into sub-sections (e.g., for extraction, traditional and modern techniques, etc.).
Figure 1 should be simplified and prepared to be more visible.
Section 8 should provide content primarily related to cereal β-D-glucans or discuss β-D-glucans from different sources in a separate section.
Other points:
What are the safety advantages of cereal β-D-glucans compared to other sources?
Regarding adding value and the health benefits of cereal β-D-glucans, add data related to estimated intakes of cereal β-D-glucans in modern diets and mention actual dietary recommendations related to their intake.
Minor changes are needed.
Author Response
Dear reviewer, the authors team would like to thank you very much for your time and energy to read carefully the manuscript and to give us usefull remarks. We tried our best to improve the quality of the paper according to your suggestions. We accepted all your comments. Thank you very much.
